# External Hemipelvectomy in Soft Tissue Sarcomas: Are They Still Needed?

**DOI:** 10.3390/cancers16223828

**Published:** 2024-11-14

**Authors:** Luis Rafael Ramos Pascua, Paula Casas Ramos, Lidia De la Cruz Gutiérrez, Maximiliano Eugenio Negri, Elena Vilar González, Julio César Córdova Peralta, María Inmaculada Mora Fernández, Jesús Enrique Vilá y Rico

**Affiliations:** 1Department of Orthopedic Surgery, Hospital Universitario 12 de Octubre, 28041 Madrid, Spain; maximilianonegri@gmail.com (M.E.N.); jccordovap@gmail.com (J.C.C.P.); vilayrico@gmail.com (J.E.V.y.R.); 2Department of Orthopedic Surgery, Hospital Universitario de León, 24071 León, Spain; casaspaula@hotmail.com (P.C.R.); lidia_delacruzg@hotmail.com (L.D.l.C.G.); elevilarg@gmail.com (E.V.G.); 3Hospital Universitario de Burgos, 09006 Burgos, Spain; maria.mora.fdez@gmail.com

**Keywords:** external hemipelvectomy, amputation, soft tissue sarcoma, pelvis, pelvic girdle, pregnancy, delivery, abdominoperineal amputation, colostomy

## Abstract

**Simple Summary:**

External hemipelvectomy as local treatment for soft tissue sarcomas is a technically demanding, mutilating surgical procedure with substantial morbidity and mortality and currently appears obsolete. This paper reviews the scientific literature on its results and current indications, with exceptional examples that illustrate and elucidate the topic.

**Abstract:**

Background: The development of new technologies, the interpretation of amputations as therapeutic failures by society, and the high morbidity and mortality associated with external hemipelvectomies make these mutilating surgical procedures appear obsolete. Herein, we review the scientific literature on the topic and present two cases of high-grade ulcerated soft tissue sarcomas in the gluteal region which show exceptional behavior and different outcomes. Methods: We performed a literature review of the PubMed databases from 2014 to April 2024. Additionally, we present two cases of soft tissue sarcomas in an 18-year-old female patient and in a 71-year-old female patient, which were treated with extended external hemipelvectomies with anterior flap, in combination with an abdominoperineal amputation and a colostomy in one case. Results: After 4 years of follow-up, case 1 is living a relatively normal life. She had an uncomplicated pregnancy and a cesarean section delivery. Case 2 underwent emergency surgery for intestinal perforation and sepsis. She died 2.5 months following the surgery. Conclusions: External hemipelvectomy for soft tissue sarcoma treatment is a demanding surgical procedure with purpose in selected cases after review by multidisciplinary committees and with informed patient consent. This should be similarly individualized and extended to other pathologies when possible.

## 1. Introduction

Hemipelvectomy is a surgical procedure involving pelvic resection, indicated for tumors in this anatomical location. It can be further classified as internal (with limb preservation) or external (with limb amputation). Both types have advantages and disadvantages, as well as technical difficulties and complications, due to the complex regional anatomy of the pelvis.

Resection in an internal hemipelvectomy can affect zones I–IV of the pelvis [1], sometimes in a combined manner. Bone reconstruction may or may not be necessary, especially after the resection of zones I and III. Although decided individually for each patient, options include iliofemoral pseudarthrosis or arthrodesis, using different grafts (allograft, free vascularized fibular graft, vascularized fibular autograft, or autoclaved autograft), or prosthetic reconstruction. Customized prosthetic reconstruction, supported by new technologies, is currently preferred, though this remains controversial [2,3,4,5,6].

The pelvic structures necessary for lower limb function include the sciatic nerve, the femoral neurovascular bundle, and the periacetabular region of the pelvis. If tumor resection requires the removal of two of these three critical anatomic structures, the patient will have a nonfunctional limb and an external hemipelvectomy (EHP) would be recommended [7]. This can be performed through the ilium (modified or conservative), the sacroiliac joint (standard or classic), or the sacral foramina (extended) [8,9,10]. Soft tissue coverage can be achieved using a posterior (gluteal) or an anterior flap, although other myocutaneous flaps can also be used, including rectus abdominis flaps, ipsilateral external oblique flaps [11], and flaps from the amputated limb itself [12].

The incidence of limb-sparing procedures performed for treating musculoskeletal tumors is increasing, facilitated by the development of new technologies. Moreover, current society interprets mutilating procedures as therapeutic failures. For both of these reasons, in addition to its inherent morbidity and mortality, external hemipelvectomy seems to be interpreted as an obsolete technique, as already stated in some past articles [13]. This study intends to revisit external hemipelvectomy in the context of soft tissue sarcomas (STSs), providing an updated literature review and two case reports that demonstrate unique presentations and behaviors.

## 2. Material and Methods

We conducted a literature search of the PubMed database and Cochrane review from 2004 to 30 April 2024. Our search strategy utilized the following terms: (“sarcoma” [MeSH Terms] OR “Soft Tissue Neoplasms” [MeSH] OR “sarcoma” OR “soft tissue sarcoma*” OR “soft-tissue sarcoma*”) AND (“hemipelvectomy*” OR “hindquarter amputation*”). Additionally, references of included studies were manually assessed to identify possible studies of interest that were not captured in the search. We reviewed the sample of studies and compared the outcomes of external hemipelvectomies, particularly in comparison to internal hemipelvectomies.

### 2.1. Case 1

A 20-year-old woman with no relevant medical history visited the emergency department of our hospital. She indicated that she had undergone surgery 4 months ago in her native country (Morocco) for a tumor in the right gluteal region, classified in the report she had provided as dermatofibrosarcoma protuberans. Physical examination revealed a vegetative, ulcerated, bleeding lesion in the same region (Figure 1A). Magnetic resonance imaging (MRI) identified a large, subcutaneous soft tissue mass measuring 15 × 12 cm in its craniocaudal and transverse planes, with multiple lobulations infiltrating the gluteal muscles, extensively contacting the iliac blade without infiltrating it, and extending beyond the median sacral crest (Figure 1B–D). A core needle biopsy was performed, and AJCC stage IIIB pleomorphic spindle cell sarcoma with negative extension status was diagnosed.

After multidisciplinary discussion, preoperative embolization of the gluteal artery (Figure 2) was performed on 2 January 2020, followed by an anterior flap extended external hemipelvectomy (Figure 3). The diagnosis was confirmed with clear resection margins, and adjuvant chemotherapy was administered.

### 2.2. Case 2

A 71-year-old woman with a history of arterial hypertension, dyslipidemia, hypothyroidism being treated with medication, mild chronic renal failure, asymptomatic dissecting aortic aneurysm (incidental finding), and a melanoma of the back that had undergone surgery 4 years earlier was transferred to our hospital on 19 November 2018. The patient was transferred from another center due to a tumor in her right gluteal region that forced her to remain bedridden in a prone position. The lesion was biopsied, and she was diagnosed with AJCC stage IIIB high-grade soft tissue sarcoma. The patient reported that the tumor had appeared 3 months earlier and had rapidly grown since then. Physical examination revealed a large, ulcerated, bleeding, vegetative lesion (Figure 4). MRI showed a lesion measuring 17 × 12 × 13 cm and infiltrating the skin deep into the gluteus maximus muscle. Multiple disseminated satellite nodules were found occupying the fat planes of the posterior thigh, the ischioanal fossa, the greater sciatic notch, and the internal obturator region. The anteromedial tumor margin was adherent and partially infiltrated the external anal sphincter and anal orifice (Figure 5). Extension status eliminated distant disease.

With a low level of awareness of the multifactorial etiology (attributed to the medication, catabolic condition, and onset of secondary polymicrobial sepsis caused by *Streptococcus agalactiae*, *S. epidermidis*, and *Ruminococcus*), a multidisciplinary discussion of the case was held. The patient’s general condition improved, and an urgent intervention was performed on 22 November 2018. An abdominoperineal resection with colostomy was performed (confirming perforation of the rectal ampulla) in combination with an anterior flap extended external hemipelvectomy (Figure 6).

## 3. Results

A literature search of the last 10 years found 124 articles in PubMed and no results in Cochrane Library related to the subject. We identified and analyzed publications on clinical cases [14,15,16,17,18,19,20,21,22,23,24], case series [2,3,4,12,25,26,27,28,29,30,31,32,33,34,35,36,37,38,39,40,41,42,43,44,45,46], meta-analyses and literature reviews [47,48], updating articles [7,49,50], publications on surgical technique [51,52,53], experimental studies [54,55,56,57], and professional biographies [58], among other items of no surgical interest. Although most publications focused on internal hemipelvectomies [2,3,14,17,18,19,26,30,34,46,49,50,57,59,60] or alternatives to hemipelvectomy [20], the reviewed articles indicate that external hemipelvectomy remains a current procedure, even in the pediatric population [32,61] (Table 1).

### 3.1. Case 1

The patient’s immediate postoperative period was uneventful. She underwent a rehabilitation program and adapted to an external prosthesis for walking. A spontaneous pregnancy occurred 3 years and 6 months post-surgery. Relative rest was recommended from week 20, and absolute rest from week 28. The pregnancy and cesarean section conducted at 37 + 1 weeks of gestation were uneventful. The cesarean section was selected to protect the pelvic floor since the birth canal was assumed to be unstructured, although vaginal delivery was not contraindicated. Furthermore, fetal development and the puerperal course were normal (Figure 7). After more than 4 years of follow-up, the patient currently shows no signs of local recurrence or distant disease. She experiences lumbar discomfort after prolonged standing and walks with the external orthosis and, occasionally, without it, with the aid of two crutches (Figure 8).

### 3.2. Case 2

The patient developed metabolic acidosis, third spacing, and anasarca in the immediate postoperative period, with active bleeding from the colostomy. An abdominal computed tomography scan revealed active arterial bleeding in the cecum in the context of ischemic colitis, which was shortly followed by local recurrence of the sarcoma and lung metastasis (Figure 9). Due to hemodynamic instability and a fluctuating confusional state, supportive care was initiated and persisted until the patient’s death on 2 February 2019.

## 4. Discussion

Currently, amputations performed in orthopedic surgical oncology are often perceived by society and a large part of the medical community as a sign of treatment failure. This may be the case when it is caused by a complication from a previous surgery, but not when it is selected as the primary treatment for an aggressive or malignant tumor that cannot be resected with clear margins, or when resection is followed by a dysfunctional reconstruction. Compared with limb-conserving surgery, amputations are performed in approximately 10% of patients with sarcoma [62]. These previous considerations are also applicable to hemipelvectomy, a procedure first performed by Billroth in 1891, although the patient survived only for a few hours. The first successful hemipelvectomy was performed in 1895 by Girard [63].

In conservative surgery of soft tissue sarcomas of the pelvic girdle, preservation of the major neurovascular structures is a necessary condition, although resection of one or both main nerves does not represent an absolute contraindication for a conservative procedure [13]. On the other hand, an ulcerated tumor would be [40], as it was in the two cases presented in our study. Except for local recurrence after internal hemipelvectomy and palliative indications, external hemipelvectomy remains a priori a curative surgical procedure. Laitinen et al. recalled that in the modern era, the main indication for major pelvic resections, including hindquarter amputation, is for local control of malignant tumors [48]. In this context, when treating malignant musculoskeletal tumors, the priority is to preserve life, then limb, and, lastly, limb function.

The most common soft tissue sarcomas of the pelvic girdle are undifferentiated pleomorphic sarcoma and liposarcoma. Local surgical treatment is guided by the same principles as those of soft tissue sarcomas in other anatomical locations, albeit with the constraints of the previously described difficulties due to regional anatomy. Moreover, similar to bone sarcomas, most are diagnosed at a larger size compared with those usually found in other, more accessible locations for physical examination. Furthermore, frequent involvement of soft tissues essential for coverage in internal hemipelvectomy compromises limb-sparing surgery.

Comparatively, and without distinguishing between bone sarcomas and soft tissue sarcomas, external and internal hemipelvectomies generally do not show significant differences regarding complications, with an overall mortality rate of ≤9% and a postoperative complication rate of 20–75% [4,39,40,64]. The most frequent complications are surgical wound infections and flap necrosis, especially in anterior flaps [13,39,64]. Among all other possible complications, urogenital trauma is estimated to occur in 1.8–2.9% of cases [4] and total complications in this anatomical region are between 8 and 18% [39]. Guder et al. [39] compared 13 internal hemipelvectomies vs. 21 external hemipelvectomies (9 as a primary procedure). Of the former, 69% required surgical revisions, with a mean number of revisions per patient of 4.1. In the EHP group, the figures were 43% and 1.8, respectively. In Couto’s series [38], there were no significant differences in perioperative data between external and internal hemipelvectomy (without reconstruction), including time to hospital discharge. Although there were also no significant differences when comparing surgical wound complications and the incidence of chronic pain, external hemipelvectomy was responsible for eight of the nine cases of surgical wound complications. The number of patients with intraoperative hemodynamic instability was higher among patients who underwent EHP than among patients who underwent internal hemipelvectomy.

Flap necrosis treatment varies. However, it can be prevented by including the gluteus maximus muscle in posterior flaps, and the rectus femoris and the vastus intermedius muscles in anterior flaps [13]. Infections occur in 10–70% of patients due to different reasons [4,39,45]. Guder et al. [39] compared internal and external hemipelvectomies and reported that complications are more frequent in the former, especially in cases of reconstruction, as well as in type II and III resections, possibly due to the proximity and sacrifice of lymphatic vessels. Another case series showed that reconstructive procedures, while maintaining joint stability, are associated with more complications [4,39,47]. A 2022 meta-analysis encourages the use of internal hemipelvectomy in pelvic sarcoma treatment, due to the lower risk of surgical site infections [47].

Regarding oncological outcomes, survival after a potentially curative hemipelvectomy depends on several factors, including the tumor histology and size, disease stage, the patient’s general condition, and the resection type [39,65]. Whether the hemipelvectomy was internal or external is not as important [47,66]. However, mean survival in these patients is highly variable. If the patient is disease-free 5 years post-surgery, sarcoma-induced mortality risk is low [67]. The same applies to local recurrences, except for bone chondrosarcoma, where late recurrences are more frequent.

Functional outcomes are particularly important following any type of hemipelvectomy. Regarding quality of life, no differences have been reported between internal and external hemipelvectomies, although those who have undergone external hemipelvectomy report more phantom limb pain. Beck et al. reported that few of the patients who underwent hemipelvectomy were independent at discharge, although most were independent 6 years later with respect to daily living activities [64]. However, long-term studies have also reported that limb function deteriorates after internal hemipelvectomy, decreasing by 23% over a follow-up period of 23–38 years [67].

External hemipelvectomy often results in better mobility at discharge, although it subsequently causes more pain, increased bladder dysfunction, and difficulty climbing stairs in half of patients [40,64]. The same study showed that 4.4% of patients walked without external aids, 81% used crutches, 9% used wheelchairs, and 6% remained bedridden [64]. Regarding the use of prostheses, most patients did not need them as they moved better without them, limiting their use for standing and for cosmetic reasons. Of the four surviving patients who underwent external hemipelvectomy in a case series by Guder et al., two used wheelchairs (one for a short time) and the other two used crutches for short distances and wheelchairs for long distances [39].

Emotional outcomes after hemipelvectomy have been poorly investigated in the scientific literature and are often limited to the corresponding section of the Musculoskeletal Tumor Society scoring system (MSTS). Finally, financial implications are difficult to calculate accurately, since social costs due to various disabilities, prosthetic changes, and those affecting the overall quality of life should all be included [68].

From the foregoing, it follows that external hemipelvectomy would be indicated as primary or secondary amputation in advanced soft tissue sarcomas when limb-preserving strategies have been exhausted, usually when neurovascular structures could not be spared due to the intrapelvic extent of the lesion [42], or in large, ulcerated lesions. Some patients undergoing potentially curative amputations can achieve long-term survival [43]. Other times, the objective will be exclusively palliative in order to contribute to the functional improvement of a very limited patient [22]. However, palliative hindquarter amputation should be critically evaluated considering its high 30-day mortality rate [36]. In this context, prospective studies on patients’ quality of life are needed to validate hemipelvectomy, taking into account their wishes, the local tumor conditions, and the previous treatments carried out [45].

Technically, soft tissue coverage in external hemipelvectomies can be performed with local flaps (gluteal or anterior myocutanoeus flap from the thigh) [45], free latissimus dorsi musculocutaneous flap [52], free fillet flap from the amputated lower limb [12,37,44,51,53], etc. Regarding the role of new technologies in surgical planning, 3D printing and computer navigation are feasible and effective procedures in pelvic tumor surgery, used to achieve the oncological aim of negative margins with the preservation of critical structures. However, their usefulness and use have been limited to date to internal hemipelvectomies [59,60].

The two cases in our study exemplify the two extremes of the possible outcomes of an external hemipelvectomy: one good oncological and functional outcome and one poor outcome with multiple complications. Both serve to ensure that patients are properly informed prior to the procedure. Our first patient should be further discussed with respect to her pregnancy and subsequent delivery.

Only 14 deliveries after hemipelvectomy had been published in the literature until 2008 [69], although other previous articles had reported a few isolated cases [70,71,72,73]. Only two new cases have been published since then [63,74]. Although most tumors were bone tumors and the flap type was unspecified, vaginal delivery remains possible. However, hemipelvectomy is associated with an increased risk of fetal malposition due to the altered pelvic anatomy and a cesarean section should be considered. In any case, a gynecological assessment is essential.

## 5. Conclusions

External hemipelvectomy with curative purposes is a surgical procedure that is still being performed in specific cases of soft tissue sarcoma, after a multidisciplinary committee case review and with patient consent. Overall, the results seem better in pelvic bone tumors [36] and in younger patients [38].

## Figures and Tables

**Figure 1 cancers-16-03828-f001:**
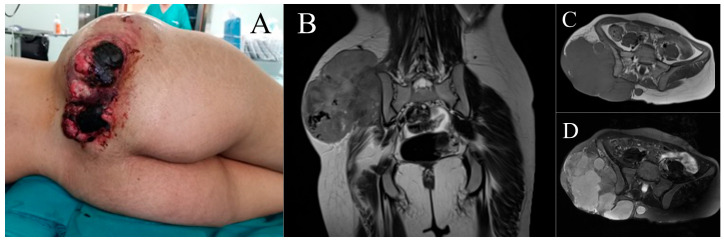
Case 1. Clinical appearance (**A**) and coronal and axial magnetic resonance images (**B**–**D**).

**Figure 2 cancers-16-03828-f002:**
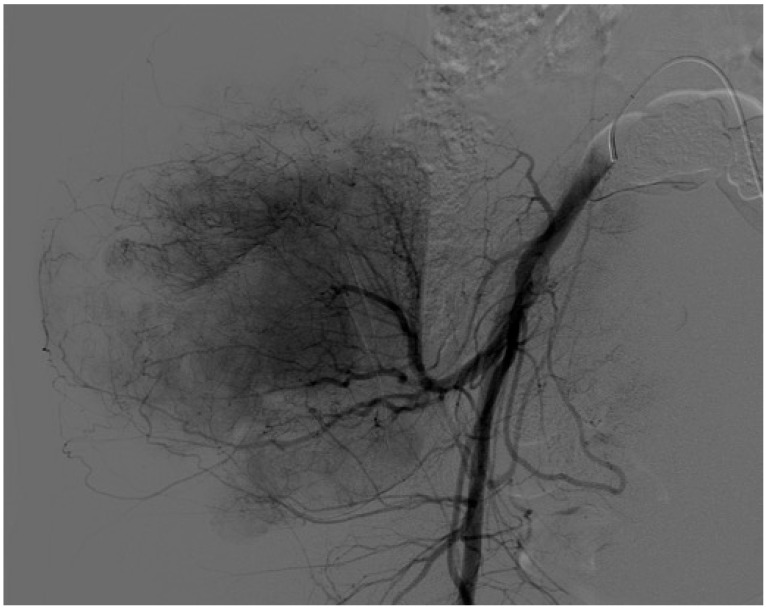
Case 1. Control arteriography for preoperative embolization.

**Figure 3 cancers-16-03828-f003:**
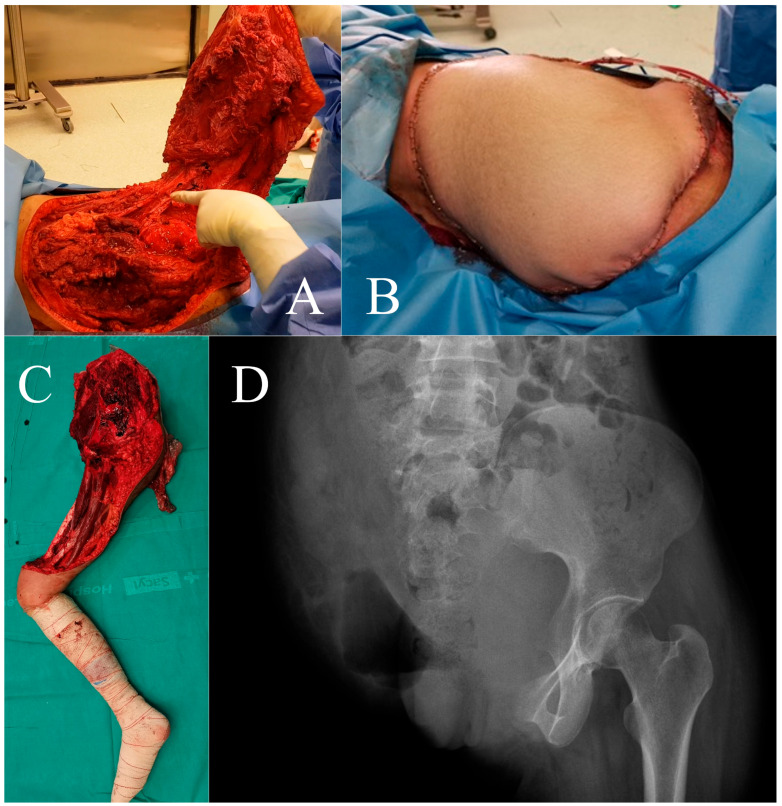
Case 1. Surgical details of anterior flap hemipelvectomy, showing the femoral neurovascular bundle (**A**) and final coverage aspect (**B**). Amputation specimen (**C**) and postoperative radiography (**D**).

**Figure 4 cancers-16-03828-f004:**
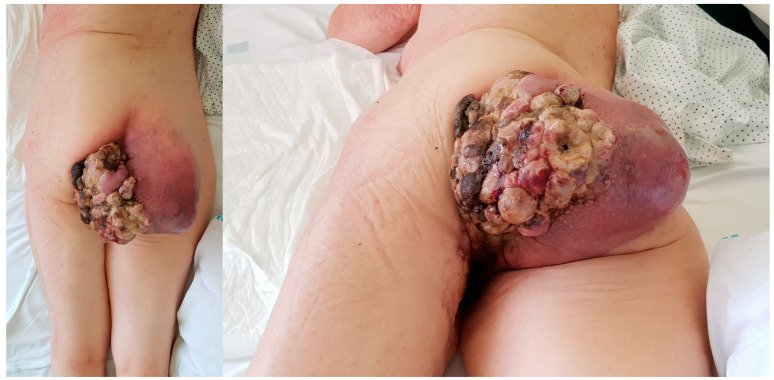
Case 2. Clinical appearance of the patient.

**Figure 5 cancers-16-03828-f005:**
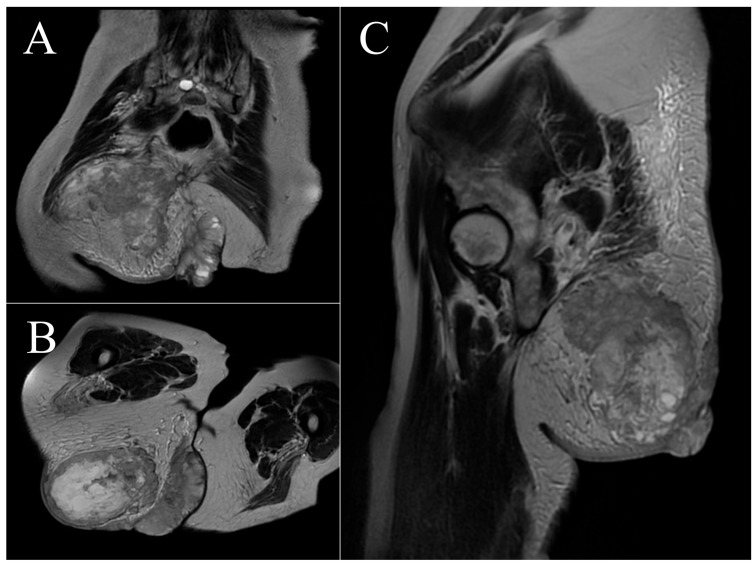
Case 2. Coronal (**A**), axial (**B**), and sagittal (**C**) magnetic resonance images.

**Figure 6 cancers-16-03828-f006:**
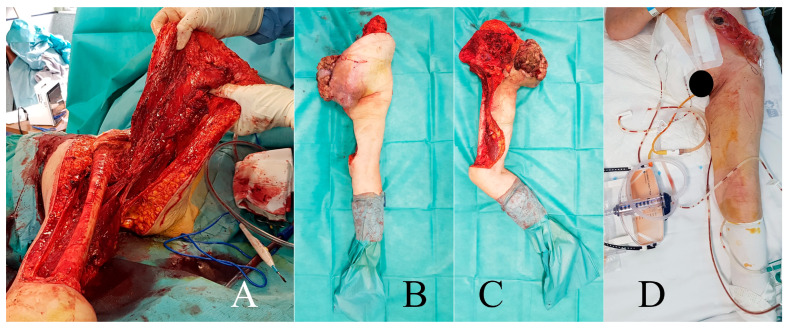
Case 2. Surgical details of anterior flap hemipelvectomy (**A**), amputation specimen (**B**,**C**), and final clinical aspect with the associated colostomy (**D**).

**Figure 7 cancers-16-03828-f007:**
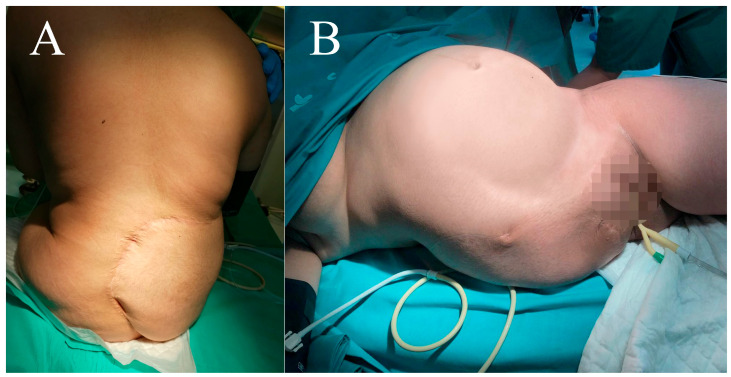
Case 1. Clinical appearance of the patient at the time of delivery. Clinical appearance of the patient at the time of delivery. Before anesthesia (**A**) and positioned on the delivery table (**B**).

**Figure 8 cancers-16-03828-f008:**
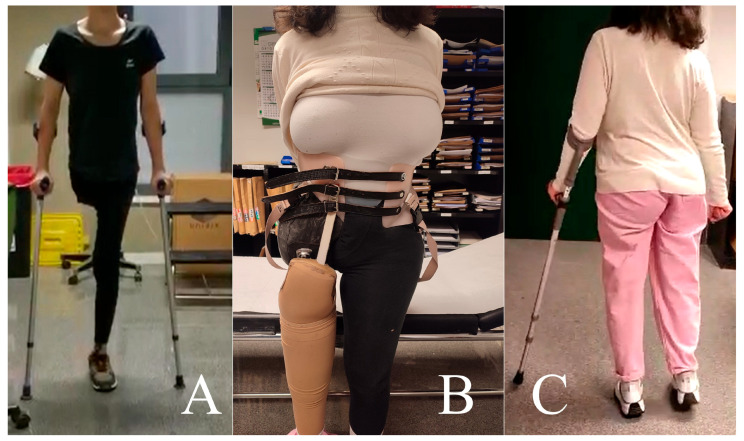
Case 1. Clinical appearance of the patient at different time points of functional recovery. Clinical appearance of the patient at different time point of functional recovery. Without external prosthesis, using two canes (**A**), and with prosthesis (**B**,**C**).

**Figure 9 cancers-16-03828-f009:**
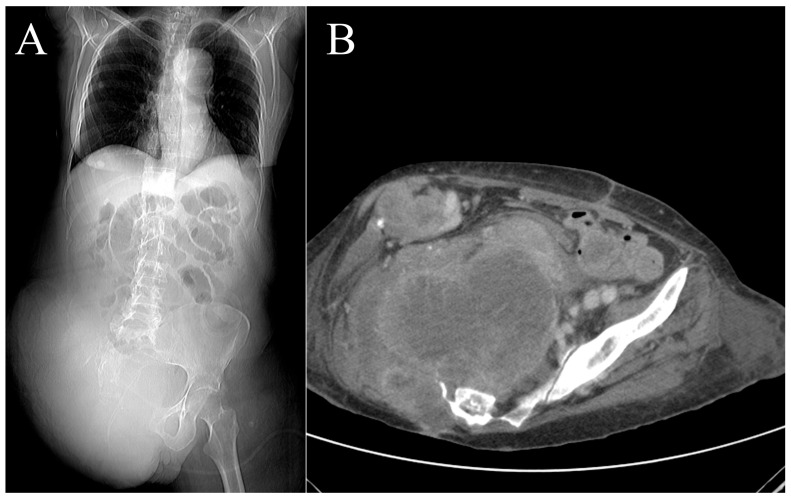
Case 2. Postoperative computed tomography scan on the patient showing the extended hemipelvectomy and tumor recurrence. Coronal (**A**) and axial (**B**) view.

**Table 1 cancers-16-03828-t001:** Summary of primary external hemipelvectomy series, with epidemiologic data and global, functional, and oncologic results.

Authors, Year	n	Mean Age (Years)	Diagnosis	Surgical Complications	Functional Results	Recurrences	Survival/Death
Puri et al., 2014 [46]	2	19.5	Osteosarcoma	1 infection	NA	2 (100%)	0% alive (FU 11.5 m)
Guder et al., 2015 [39]	9	70.7	BT. STS	9 surgical revisions ^a^	NA	3 (14.3%) ^a^	16 (76.2%) ^a^ died
Freitas et al., 2015 [45]	8	NA	BT. STS	NA	NA	NA	NA
Read et al., 2015 [45]	3	NA	Melanoma	NA	1 alive (use crutches)	NA	- 2 died (<1 year) - 1 alive (7 years)
Furtado et al., 2015 [41]	22	NA	BTS/STS	NA	- 4.8% use prosthetic limb at least daily - 83.3% use walking aids	NA	NA
Couto et al., 2016 [40]	23	45	BT. STS	NA	NA	NA	11% alive (5 years)
Puchner et al., 2017 [42]	19	NA	BTS	NA	NA	NA	NA
van Houdt et al., 2018 [38]	78	NA	BTS/STS	70% at leats one	50% of the patients survivors > 1 year performed reasonably well	8 (10.3%)	- Palliative goal: MS 5 months - Curative goal: 30 months
Roulet et al., 2018 [12]	6	53.6	BTS/STS	6	NA	NA	- 4 died - 2 alive (MS 7.5 months). - 25% alive at 10 years
Tashiro et al., 2019 [52]	6	65	BT. STT	3 infections	- 3 wheel chair - 3 Lower limb prosthesis	NA	4 alive (FU not specified)
Kiiski et al., 2020 [36]	136 ^b^	51	MBT/STS	73 (53.7%) at least one	NA	25.3% BTS/28.9% STS	59.3 ^c^/12.5 ^d^ (MS)
Kiiski et al., 2021 [37]	12	60	BTS/STS	50%	- 4 NA - 8 crutches and/or wheel chair	1 (8.3%)	58% (3 years)
Kreutz-Rodrigues et al., 2020 [44]	7	53.1	BTS/STS	1 wound dehiscence	NA	2 (28.6%)	4 alive (FU not specified)
Karaka et al., 2022 [4]	29	43	Tumor. Hydatid cyst	20 infections (8 superficial, 12 deep)	NA	NA	25.7 months (MS)

Abbreviations: BT = bone tumor, STS = soft tissue sarcoma, BTS = bone tumor sarcoma, MBT = malignant bone tumor, HP = hemipelvectomy, NA = not available, MS = median survival, FU = follow-up. ^a^ Calculated on 21 hemipelvectomies (external and internal). ^b^ Primary and secondary HP. ^c^ Median survival with primary HP in STS. ^d^ Median survival with secondary HP in STS. The table has limitations inherent to the topic addressed, with publications of clinical cases involving different pathologies and, sometimes, without specifying the specific results of the technique. Cases of internal hemipelvectomy in comparative studies have been excluded.

## Data Availability

The datasets used and/or analyzed during the current study are available from the corresponding author upon reasonable request.

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
