# Peer review of "External Hemipelvectomy in Soft Tissue Sarcomas: Are They Still Needed?"

_cancers, 2024, doi:10.3390/cancers16223828_

Round 1

Reviewer 1 Report

Comments and Suggestions for Authors

This review of the Literature summarize the role of hindquarter amputation in soft tissue sarcomas.

The topic is interesting. However, the paper looks more like a case report rather than a review of the Literature. 

The review of the Literature should include more than one database. Also, please extend the time window for Literature search.

Wouldn't define amputation "obsolete"

Review of the Literature is not reported. Please resume it in a table. 

Discussion is poorly presented and not based on results. In my opinion, the paper would benefit from indications to hindquarter amputations in STS. When is it necessary? Actually, from the cases presentation it is not clear why  hindquarter amputation was required. Were vessels or nerves involved?

Comments on the Quality of English Language

Many grammar and syntax errors.

Author Response

Thank you very much for reviews. Answers in red and modifications highlighted in yellow.

  • The topic is interesting. However, the paper looks more like a case report rather than a review of the Literature. 

It certainly seems so, but the clinical cases have only been presented as exceptional illustrative examples of the main body of the paper, in order to give it relevance. The goal of the paper is a review of the Literature on external hemipelvectomy in soft tissue sarcomas, a topic little studied at present.

  • The review of the Literature should include more than one database. Also, please extend the time window for Literature search.

We have performed a search with the same search strategy on Cochrane Library with no results. For this reason we considered it enough to limit the search to the Medline/Pubmed database, as other authors have done (Hope WC, Ferro LC, Snyder JA, Procter LD, Salluzzo JL. Hemipelvectomy hernia: case series and literature review. Hernia. 2021 Oct;25(5):1159-1167). However, we have extended the comprehensive literature search to 10 years (from 2014 to 2024).

We expand on this information in the corresponding sections of the article: Material and methods and results.

  • Wouldn't define amputation "obsolete"

Of course, amputation is not an obsolete technique. We wanted to refer to the interpretation of hemipelvectomy in the collective consciousness of society. We changed the meaning of the phrase like this:

SAID: For both these reasons, in addition to its inherent morbidity and mortality, external hemipelvectomy seems to have become an obsolete technique, as already stated in some classic articles [13].

SHOULD SAY: For both these reasons, in addition to its inherent morbidity and mortality, external hemipelvectomy seems to be interpreted as an obsolete technique, as already stated in some classic articles [13].

  • Review of the Literature is not reported. Please resume it in a table. 

Agreed. We add a table summarising the Literature.

Table 1: Summary of primary external hemipelvectomies series, with epidemiologic data and global functional and oncologic results. Abbreviations: BT=bone tumor, STS=soft tissue sarcoma, BTS=bone tumor sarcoma, MBT=malignant bone tumor, HP=hemipelvectomy, NA=Not available, MS=Median survival, FU=Follow-up, aCalculated on 21 hemipelvectomies (externals and internals). bPrimary and secondary HP. cMedian survival with primary HP in STS. dMedian survial with secondary HP in STS. The table has limitations inherent to the topic addressed, with publications of clinical cases, of different pathologies and sometimes without specifying the specific results of the technique. Cases of internal hemipelvectomies in comparative studies have been excluded.

We have expanded the literature review to 10 years (from 2014 to 2024). For the same reason we have extended the number of references to 78.

  • Discussion is poorly presented and not based on results. In my opinion, the paper would benefit from indications to hindquarter amputations in STS. When is it necessary? Actually, from the cases presentation it is not clear why  hindquarter amputation was required. Were vessels or nerves involved?

We added an extensive paragraph at the end of the discussion to respond to the current indications for external hemipelvectomy, with basic technical notes. However, it is already stated in the introduction that if tumor resection requires removal of two of these three critical anatomic structures, the patient will have a nonfunctional limb and an external hemipelvectomy (EHP) would be recommended [7]. We also modified the conclusion for a better and clearer understanding of the subject.

From the foregoing, it follows that external hemipelvectomy would be indicated as primary or secondary amputation in advanced soft tissue sarcomas when limb-preserving strategies have been exhausted, usually when neurovascular structures could not be spared due to the intrapelvic extent of  the lesion [43]; or in large ulcerated lesions. Some patients undergoing potentially curative amputations can achieve long-term survivals [44]. Other times, the objective will be exclusively palliative in order to contribute to the functional improvement of a very limited patient [22]. However, palliative hindquarter amputation should be critically evaluated considering high 30-day mortality rate [36]. In this context, prospective studies on patients' quality of life are needed to validate hemipelvectomy, taking into account their wishes, local tumour conditions and previous treatments carried out [46].

Technically, soft tissue coverage in external hemipelvectomies can be performed with local flaps (gluteal or anterior myocutanoeus flap from the thigh) [46], free latissimus dorsi musculocutaneous flap [53], free fillet flap from the amputated lower limb [37,40,45,52,54], etc. Regarding the role of new technologies in surgical planning, 3D-printing and computer navigation are procedures feasibles and effectives in pelvic tumor surgery to achieve the oncological aim of negative margins with preservation of critical structures. However, its usefulness and use has been limited to date to internal hemipelvectomies [60,61].

The general paragraph on complications has also been completed in the discussion:

Guder et al [39] compared 13 internal hemipelvectomies vs 21 external hemipelvectomies (9 as a primary procedure). Of the former, 69% required surgical revisions with a mean number of revisions per patient of 4.1. In the EHP group the figures were 43% and 1.8 respectively. In the Couto´s series [38] there were no significant differences in perioperative data between external and internal hemipelvectomy (without reconstruction), including time to hospital discharge. Although there were also no significant differences when comparing surgical wound complications and the incidence of chronic pain, external hemipelvectomy was responsible for 8 of the 9 cases of surgical wound complications. The number of patients with intraoperative hemodynamic instability was higher among patients who underwent EHP than among patients who underwent internal hemipelvectomy.

  • Comments on the Quality of English Language: Many grammar and syntax errors.

Sorry. The English translation was reviewed by a professional. We suggest resubmitting the final version of the manuscript, if approved, for correction.

Reviewer 2 Report

Comments and Suggestions for Authors

Thank you for addressing this important issue in sarcoma surgery. I completely agree with the authors that amputation is nowadays often judged as a failure of treatment. However, there is a small subgroup of patients which would die from their local recurrence/tumor and could be very well treated by hemipelvectomy.

The two cases presented by the authors are illustrative, however I think their literature search could be presented much better: First suggestion is to broaden the range of the search: why 2022 until 2024 is chosen? I think it would be much more interesting to look at least for the last ten years. Moreover, I would suggest to summarize the results of the literature search in a table in the results part of the manuscript.

In the discussion part I would like to find a comprehensive recommendation for surgeons and orthopedic surgeons that treat sarcoma patients: when and in which cases to go for hemipelvectomy. And maybe also add the technical conditions (which structures should be uninvolved by tumor).

In my opinion figure 7 does not add anything to the manuscript and should be removed in a next version.

Comments on the Quality of English Language

English language seems fine.

Author Response

TO REVIEWER 2

Thank you very much for reviews. Answers in red and modifications highlighted in yellow.

  • Thank you for addressing this important issue in sarcoma surgery. I completely agree with the authors that amputation is nowadays often judged as a failure of treatment. However, there is a small subgroup of patients which would die from their local recurrence/tumor and could be very well treated by hemipelvectomy.

Thank you. We share this appreciation                                 

  • The two cases presented by the authors are illustrative, however I think their literature search could be presented much better: First suggestion is to broaden the range of the search: why 2022 until 2024 is chosen? I think it would be much more interesting to look at least for the last ten years. Moreover, I would suggest to summarize the results of the literature search in a table in the results part of the manuscript.

Agreed. We have expanded the literature review to 10 years (from 2014 to 2024) and added a table summarising the main series. For the same reason we have extended the number of references to 78.

We add a table summarising the Literature.

Table 1: Summary of primary external hemipelvectomies series, with epidemiologic data and global functional and oncologic results. Abbreviations: BT=bone tumor, STS=soft tissue sarcoma, BTS=bone tumor sarcoma, MBT=malignant bone tumor, HP=hemipelvectomy, NA=Not available, MS=Median survival, FU=Follow-up, aCalculated on 21 hemipelvectomies (externals and internals). bPrimary and secondary HP. cMedian survival with primary HP in STS. dMedian survial with secondary HP in STS. The table has limitations inherent to the topic addressed, with publications of clinical cases, of different pathologies and sometimes without specifying the specific results of the technique. Cases of internal hemipelvectomies in comparative studies have been excluded.

  • In the discussion part I would like to find a comprehensive recommendation for surgeons and orthopedic surgeons that treat sarcoma patients: when and in which cases to go for hemipelvectomy. And maybe also add the technical conditions (which structures should be uninvolved by tumor).

We added an extensive paragraph at the end of the discussion to respond to the current indications for external hemipelvectomy, with basic technical notes. We also modified the conclusion for a better and clearer understanding of the subject.

From the foregoing, it follows that external hemipelvectomy would be indicated as primary or secondary amputation in advanced soft tissue sarcomas when limb-preserving strategies have been exhausted, usually when neurovascular structures could not be spared due to the intrapelvic extent of  the lesion [43]; or in large ulcerated lesions. Some patients undergoing potentially curative amputations can achieve long-term survivals [44]. Other times, the objective will be exclusively palliative in order to contribute to the functional improvement of a very limited patient [22]. However, palliative hindquarter amputation should be critically evaluated considering high 30-day mortality rate [36]. In this context, prospective studies on patients' quality of life are needed to validate hemipelvectomy, taking into account their wishes, local tumour conditions and previous treatments carried out [46].

Technically, soft tissue coverage in external hemipelvectomies can be performed with local flaps (gluteal or anterior myocutanoeus flap from the thigh) [46], free latissimus dorsi musculocutaneous flap [53], free fillet flap from the amputated lower limb [37,40,45,52,54], etc. Regarding the role of new technologies in surgical planning, 3D-printing and computer navigation are procedures feasibles and effectives in pelvic tumor surgery to achieve the oncological aim of negative margins with preservation of critical structures. However, its usefulness and use has been limited to date to internal hemipelvectomies [60,61].

  • In my opinion figure 7 does not add anything to the manuscript and should be removed in a next version.

Completely agree. Figure 7 is removed.

  • Comments on the Quality of English Language: English language seems fine.

Ok. Thanks. The English translation was reviewed by a professional.

Reviewer 3 Report

Comments and Suggestions for Authors

Dear Editor,

 the authors reported two cases of external hemipelvectomy in soft tissue sarcomas (STS), providing an updated literature review and two case reports demonstrating unique presentations and behaviors.

However, the data procedures images lack a table to compare literature data and their findings  In my opinion, a table  could help a general audience to better understand the  variability among prognosis 

A paragraph on novel technology should be introduced

Comments on the Quality of English Language

The English needs revision some sentences in the introduction  are very complicated

Author Response

TO REVIEWER 3

Thank you very much for reviews. Answers in red and modifications highlighted in yellow.

  • The authors reported two cases of external hemipelvectomy in soft tissue sarcomas (STS), providing an updated literature review and two case reports demonstrating unique presentations and behaviors.

Yes

  • However, the data procedures images lack a table to compare literature data and their findings  In my opinion, a table  could help a general audience to better understand the  variability among prognosis 

Agreed. We add a table summarising the Literature.

Table 1: Summary of primary external hemipelvectomies series, with epidemiologic data and global functional and oncologic results. Abbreviations: BT=bone tumor, STS=soft tissue sarcoma, BTS=bone tumor sarcoma, MBT=malignant bone tumor, HP=hemipelvectomy, NA=Not available, MS=Median survival, FU=Follow-up, aCalculated on 21 hemipelvectomies (externals and internals). bPrimary and secondary HP. cMedian survival with primary HP in STS. dMedian survial with secondary HP in STS. The table has limitations inherent to the topic addressed, with publications of clinical cases, of different pathologies and sometimes without specifying the specific results of the technique. Cases of internal hemipelvectomies in comparative studies have been excluded.

We have expanded the literature review to 10 years (from 2014 to 2024). For the same reason we have extended the number of references to 78.

  • A paragraph on novel technology should be introduced

Agreed. We add a paragraph with reference to new technologies:

Technically, soft tissue coverage in external hemipelvectomies can be performed with local flaps (gluteal or anterior myocutanoeus flap from the thigh) [46], free latissimus dorsi musculocutaneous flap [53], free fillet flap from the amputated lower limb [37,40,45,52,54], etc. Regarding the role of new technologies in surgical planning, 3D-printing and computer navigation are procedures feasibles and effectives in pelvic tumor surgery to achieve the oncological aim of negative margins with preservation of critical structures. However, its usefulness and use has been limited to date to internal hemipelvectomies [60,61].

  • Comments on the Quality of English Language: The English needs revision some sentences in the introduction  are very complicated

I´m sorry. The English translation was reviewed by a profesional, but we can review the final version, if approved.

Round 2

Reviewer 1 Report

Comments and Suggestions for Authors

I appreciate the Authors' efforts in the attempt to ameliorate the paper.

They were able to address appropriately to most of my previous concerns.

Reviewer 3 Report

Comments and Suggestions for Authors

Dear Editor

the authors addressed all my points 

I agree to publication

Comments on the Quality of English Language

The English is good